

# Gender-associated factors for frailty and their impact on hospitalization and mortality among community-dwelling older adults: a cross-sectional population-based study

Qin Zhang, Huanyu Guo, Haifeng Gu and Xiaohong Zhao

Department of Geriatrics, First Affiliated Hospital, School of Medicine, Zhejiang University, Hangzhou, Zhejiang, P.R. China

## ABSTRACT

**Background**. Frailty associated with aging increases the risk of falls, disability, and death. We investigated gender-associated factors for frailty.

**Methods**. Data of 3,079 geriatric subjects were retrieved from the National Health and Nutrition Examination Survey (NHANES) 2007–2010 database. After excluding 1,126 subjects with missing data on frailty, medical history and survival, data of 1,953 patients were analyzed. Main endpoints were frailty prevalence, mortality rates and causes of death.

**Results**. Frailty prevalence was 5.4% in males, 8.8% in females. Significant risk factors for geriatric frailty in males were being widowed/divorced/separated, low daily total calorie intake, physical inactivity, sleeping >9 h, smoking and hospitalization history; and in females were obesity, physical inactivity, sleeping <6 h, family history of diabetes and heart attack, and hospitalization history. Frail subjects had higher mortality rates (22.5% male; 8.5% female) than pre-frail (8.7% male; 6.4% female) and non-frail (5.4% male; 2.5% female). Main causes of death were heart diseases (41%) and chronic lower respiratory diseases (23.0%) in males and nephritis/nephrosis (32.3%) and chronic lower respiratory diseases (17.6%) in females.

**Discussion**. Factors associated with frailty differ by gender, with higher frailty prevalence in females and higher mortality in males. Gender-associated factors for frailty identified in this study may be useful in evaluating frailty and guiding development of public health measures for prevention.

**Key Message**. Common predictive factors for frailty among older adults of both genders, including more frequent previous hospitalizations, physical inactivity, and certain gender-associated factors for frailty, are consistent with results of other NHANES studies in which self-reported higher levels of illness and sedentary behavior were directly associated with frailty.

Corresponding author
Xiaohong Zhao, 1202124@zju.edu.cn

## INTRODUCTION

Frailty in older adults results from cumulative decline in physiological functioning and increased physical and mental vulnerability associated with aging, reducing the ability of older adults to respond effectively to illness or trauma (*Walston et al., 2006*; *Eeles et al., 2012*; *Clegg et al., 2013*). Frailty in older adults increases the risk for adverse outcomes associated with falls, delirium, and incontinence; complications of treatment with medications or procedures; hospitalization and disability (*Song, Mitnitski & Rockwood, 2010*). While frailty in older adults is associated with increased symptoms and complex diagnoses, their tolerance for medical interventions is diminishing simultaneously (*Walston et al., 2006*). However, while one-quarter to one-half of adults over age 85 are estimated to be frail and experience functional decline without obvious stressors or illnesses, many older adults remain vigorous into their later years (*Clegg et al., 2013*).

Differences between frail and non-frail older adults have led to the development of various screening tools to assess frailty risk and facilitate epidemiologic study. Some of these screening tools are based on frailty models such as the phenotype model (*Fried et al., 2001*) and cumulative deficit model (*Mitnitski, Mogilner & Rockwood, 2001*), or on functional restrictions, although none of these reliably identify frailty associated with aging (*Sternberg et al., 2011*). A frailty index based on impairments of cognitive function, mood, motivation, communication, mobility and incontinence, activities of daily living, nutrition, social resources, and comorbidities, is highly predictive of institutionalization or death (*Rockwood et al., 2006*). Various frailty indicator questionnaires have also been developed, including the Frail Elderly Functional Questionnaire, which is sensitive to changes in status (*Gloth et al., 1999*). Nevertheless, while the clinical utility of these screening tools remains limited, frailty must be diagnosed to help slow the progression to disability, institutionalization, and death (*Miller et al., 2017*).

Physical inactivity is one of the strongest risk factors for frailty along with aging (*Ma et al., 2017*). Body composition in men and fat percentage in women are also associated with increased risk of frailty (*Waters et al., 2012*). The prevalence of frailty is found to be greater in women (*Waters et al., 2012*; *Ma et al., 2017*). In a European study, the frailty-free years of women were significantly fewer than those of men, influenced by both biological and socio-economic factors (*Romero-Ortuno, Fouweather & Jagger, 2014*). Although women are noted to live longer than men, their health status may be poorer due in part to environmental influences on frailty and that women are affected more than men by lifestyle factors, increasing their vulnerability to subcellular mechanisms that increase recovery time (*Hubbard, 2015*). Health and mortality were affected negatively by smoking in both men and women, and smokers were frailer than non-smokers; however, women who smoked lost their survival advantage (*Wang et al., 2013*). Differences in the frailty index for men (0.244) and women (0.278) may stem from evolutionary design as well as biological and socio-behavioral factors such as, for example, fitness frailty in men and fertility frailty in women (*Hubbard, 2015*).

We hypothesized that identifying gender-associated risks for frailty would be useful in frailty assessment and may help to address biological and lifestyle factors that contribute to

frailty. Therefore, we aimed to investigate gender-associated risk factors for geriatric frailty and their impact on hospitalization and mortality.

## PATIENTS AND METHODS

### Data source

The present study analyzed respondent data from the National Health and Nutrition Examination Survey (NHANES), which was collected in two cycles (2007–2008 & 2009–2010) by the Centers for Disease Control and Prevention (CDC), National Center for Health Statistics (NCHS) in the USA. All data were from the Public Data General Release File documents, CDC and NCHS, U.S. Department of Health and Human Services, Hyattsville, MD, USA (*Centers for Disease Control and Prevention, National Center for Health Statistics, 2008*). The data are released for research purposes and permission to use the data is granted to researchers by the NCHS.

### Ethical considerations

Ethical approval of the NHANES program and signed informed consent by participants were obtained prior to data collection by NHANES, therefore, no further ethical approval and informed consent were required for the present study. Additionally, all NHANES data released by the NCHS are de-identified and the data remain anonymous during data analysis.

### Study population

Data of 3,079 community-dwelling geriatric subjects with mean age 73.8 years and 48.6% male from two cycles of NHANES data collection during 2007–2010 were eligible for inclusion. After excluding 1,123 subjects with missing data on frailty (e.g., body weight, health status, fatigue, difficulty walking a specific distance), disease history (e.g., asthma, diabetes, cancer, osteoarthritis, heart disease, stroke, emphysema, chronic bronchitis, renal failure, etc.), and three subjects without survival data, data of 1,953 subjects were analyzed. Differences between included and excluded subjects are shown in Table S1. The included subjects were significantly younger with greater percentages of non-Hispanic White, married/living with partner, normal BMI, higher education level, higher family income/poverty ratio, and family/personal medical history of hormone replacement therapy; they also had less family/personal medical history of asthma, diabetes, heart attack, and less hospitalization <3 times, mental health consultation, osteoporosis and steroid usage.

### Main outcome measures

*Primary endpoint*: The primary endpoint of the present study was the prevalence of geriatric frailty. All included subjects were categorized into non-frail (scored 0), pre-frail (scored 1–2) and frail (scored 3–5) clusters according to the previously validated FRAIL Scale (*Morley, Malmstrom & Miller, 2012*). Pre-frail is defined as individuals at risk for frailty who fulfill some, but not all, frailty criteria. Five FRAIL scale parameters (fatigue, resistance, ambulation, illness, and weight loss) are scored from 0 to 5, representing the gradually increased presence of each parameter.

*Secondary endpoints:* The secondary endpoints were mortality rates of all subjects and the major causes of death. NHANES respondents were linked to NDI mortality data through December 31,2011 for this section of the survey report.

## Study variables

Variables recorded for each case included patient demographics (age, gender, race/ethnicity, marital status), family medical history (diabetes, asthma, heart attack/angina, osteoporosis); health and medical conditions (self-reported osteoporosis, fracture, prednisone or cortisone use, mental health consultation, hormone replacement therapy): socioeconomic status (education level, family poverty income ratio, health insurance status); lifestyle or behavioral factors (BMI, smoking history, alcohol use, milk consumption, vegetarian status, food allergy, water devices use, physical activity, sleep duration); and dietary factors (total daily calorie consumption, total daily macronutrients consumption, vitamin D insufficiency, iron deficiency). Male and female data were analyzed separately to obtain different unique risk factors for each gender. Data were obtained as follows:

- Demographic data were collected in home visits by trained interviewers using the Family and Sample Person Demographics questionnaires and Computer-Assisted Personal Interviewing (CAPI) system (Confirmit Corp., New York, NY, USA). Race/ethnicity was self-reported as Mexican American, Other Hispanic, Non-Hispanic White, Non-Hispanic Black, and Other Races, including multiracial.
- Family medical history data were self-reported using NHANES interviewer-administered questionnaires (Medical conditions).
- Socioeconomic status data were collected using the NHANES Demographic Variables and Sample Weights questionnaire. For the ratio of family income to poverty threshold, the range of values is continuous from 0 to 5; values of 5 and greater recorded as 5.
- Lifestyle and behavior factors were collected separately as follows:

*Body Mass Index (BMI)* data were based on "Body Measures" of NHANES Examination Protocol and were collected at the NHANES Mobile Examination Center (MEC) by trained healthcare technicians. BMI data were defined by World Health Organization (WHO) criteria as: underweight (BMI $< 18.5$ Kg/m$^2$), normal (BMI $= 18.5\sim24.9$ Kg/m$^2$), overweight (BMI $= 25\sim29.9$ Kg/m$^2$), obese (BMI $\geq 30.0$ Kg/m$^2$) (*National Health and Nutrition Examination Survey, 2013*).

*Smoking History* was recorded using interviewer-administered NHANES questionnaires (Smoking–Cigarettes Use). Respondents were categorized as current regular smoker or never regular smoker.

*Alcohol Use* was evaluated according to amounts determined to increase health risks by the U.S. National Institute on Alcohol Abuse and Alcoholism (NIAAA) (*National Institute on Alcohol Abuse and Alcoholism, 2005*). Alcohol consumption levels were defined by Dietary Guidelines for Americans 2010 (*US Department of Agriculture, US Department of Health and Human Services, 2010*): heavy drinkers consume more than 14 standard drinks per week on average or more than four drinks on any day for men; and more than seven standard drinks per week on average or more than three drinks on any day for women; moderate drinkers consume one drink per day for women and up

to two drinks per day for men. Occasional and non-drinkers are those who do not fall into the above criteria.

*Physical Activity* was measured by WHO Metabolic Equivalents of Task (METs) scores (ratio of working metabolic rate relative to resting metabolic rate) (*World Health Organization, 2017*). One MET is defined as the energy cost of sitting quietly and is equivalent to a calorie consumption of 1 kcal/kg/h. MET scores were calculated using interviewer-administered NHANES questionnaires (Physical Activity). The cutoff point for MET scores was established as 600 MET-min/week equal to a moderate intensity of physical activity based on WHO recommendation as described previously (*Ainsworth et al., 2011*).

*Sleep Duration* was evaluated by asking: "How much sleep do you usually get at night on weekdays or workdays?" and recording the number of hours. The *National Sleep Foundation (2017)* has updated recommendations for daily sleep amounts across the lifespan, including new ranges for each age group. Data were categorized as normal, short, and long sleep duration. For adults >65 years, sleeping 7∼8 h daily was considered appropriate and recommended.

*Other Factors* added as variables include milk consumption, food allergies, water treatment devices use and vegetarian status.

- Dietary factors were collected by in-person interviews conducted in a private room at the NHANES Mobile Examination Center (MEC). Measuring guides (e.g., various glasses, bowls, mugs, drink boxes/bottles, household spoons, measuring cups/spoons, ruler, thickness sticks, bean bags and circles) helped participants report food amounts. Detailed dietary intake data included food consumed during previous 24 h and total daily calorie & macronutrients. Data were categorized based on Dietary Guidelines for Americans, 2010, (*US Department of Agriculture, US Department of Health and Human Services, 2010*), including recommended intake and higher and lower intake by age and gender. Vitamin D insufficiency was evaluated based on serum 25-hydroxyvitamin D data from the NHANES Laboratory Data Protocol (*National Health and Nutrition Examination Survey, 2017*) and respective interpretation of deficiency levels. For the present study, the cutoff point was established as <40 nmol/L, as previously described (*Schleicher et al., 2016*). Iron deficiency was evaluated based on Standard Biochemistry Profile from the NHANES Laboratory Data Protocol (*National Health and Nutrition Examination Survey, 2017*). Normal ranges were 55–160 µg/dL for men and 40–155 µg/dL for women; deficiency was defined as lower than these ranges.

- Disease associations included medical conditions (osteoporosis, fracture, prednisone or cortisone use, female hormone replacement therapy, and mental health consultation) self-reported during interviewer-administered questionnaire data collection.

- Hospitalization utilization was obtained using the Hospital Utilization and Access to Care questionnaire from the NHANES database administered in home visits by trained interviewers using the CAPI software interview system (Confirmit Corp., New York, NY, USA). Data included frequency of overnight or longer hospital stay during past 12 months, categorized as no hospitalization, ≤3 times and more than three times.

## Statistical analysis

A multistage probability sampling design was used, employing a combined 4-year dietary day-one sample weight calculated according to the *National Center for Health Statistics (2017)* method and applied to all analyses. *T*-tests were used for continuous variables and chi-square tests were used for categorical variables. Weighted linear regression analysis was performed to test differences in continuous characteristics between participants clustered according to health status levels of non-frail, pre-frail, and frail, while logistic regression was performed for categorical parameters and was implemented to identify factors associated with health status levels. Significant variables revealed in univariate analysis were used to establish the final multivariable model. Differences and effectors of geriatric frailty were examined separately by gender. All analyses were performed based on the NHANES sampling design and appropriate weighting of participants in the statistical models. All statistical analyses were two-tailed at 0.05 significance level. All analyses were completed using SAS Version 9.3 (SAS Institute Inc., Cary, NC, USA).

## RESULTS

### Subjects' demographic and clinical characteristics

In the present study, the prevalence of frailty was 5.4% in males and 8.8% in females. Prevalence of pre-frailty ranged from 38.1% in males to 40.1% in females. In males, race, obesity, health insurance, family histories, mental health consultation and prevalence of fracture were equally distributed among the three clusters of non-frail, pre-frail and frail subjects. However, non-frail subjects had younger age, greater percentage of married/living with partner, higher family poverty/income ratio, higher education level, less presence of osteoporosis or steroid usage, and more than three times less hospitalization history compared with pre-frail and frail subjects (Table 1).

Differences were also found in macronutrients intake and lifestyle or behavioral factors between the three clusters of males, which included participants with low daily total calorie intake, low daily total protein intake, low daily total fat intake, iron deficiency, physical inactivity, shorter sleep duration, and smoking status (Table 2).

In females, the three clusters of non-frail, pre-frail and frail subjects also shared many similarities, however non-frail subjects had younger age, a greater percentage of normal BMI, more hormone replacement therapy, less family history of heart attack, less hospitalization history of over three times, less vitamin D insufficiency, less high C-reactive protein (CRP), more physical inactivity over 600 MET and shorter sleep duration compared with pre-frail and frail subjects (Tables 1 and 2).

The characteristics of subjects with missing data ($n = 1,126$) are compared with those of the study group ($n = 1,953$) according to "included" and "excluded" subjects (Table S1).

### Causes of death

Major causes of death in study subjects are shown in Table 3. A higher mortality rate (22.5%) was found in frail men compared with non-frail (5.4%) and pre-frail (8.7%) men. Among males, most subjects in the non-frail and pre-frail clusters died of malignant

**Table 1  Subjects' demographic and anthropometric characteristics and personal/family medical history by gender and frailty cluster.**

| Variables | Male (n = 1005) | | | p-value | Female (n = 948) | | | p-value |
|---|---|---|---|---|---|---|---|---|
| | Non-frail (n = 567) | Pre-frail (n = 383) | Frail (n = 55) | | Non-frail (n = 485) | Pre-frail (n = 380) | Frail (n = 83) | |
| Demographic features | | | | | | | | |
| Age, years | 71.96 ± 0.22 | 72.58 ± 0.29 | 75.38 ± 1.14 | 0.001* | 72.47 ± 0.30 | 73.66 ± 0.38 | 73.70 ± 0.77 | 0.020* |
| Race/ethnicity | | | | 0.479 | | | | 0.140 |
| Mexican American | 45(2.9) | 45(4.8) | 3(2.5) | | 48(2.9) | 49(3.9) | 4(1.5) | |
| Non-Hispanic White | 370(84.1) | 242(82.2) | 43(86.8) | | 328(87.7) | 230(83.0) | 53(84.5) | |
| Non-Hispanic Black | 87(6.3) | 62(7.7) | 5(3.8) | | 63(6.2) | 61(8.3) | 19(10.9) | |
| Other Hispanic | 44(2.7) | 24(2.5) | 2(2.0) | | 42(2.4) | 32(2.6) | 6(2.4) | |
| Other race | 21(3.9) | 10(2.8) | 2(4.8) | | 4(0.8) | 8(2.2) | 1(0.7) | |
| Marital status | | | | 0.024* | | | | 0.169 |
| Married/Living with partner | 442(82.7) | 265(73.9) | 42(79.8) | | 242(56.2) | 162(47.2) | 36(52.8) | |
| Widowed/Divorced/Separated | 103(14.0) | 108(23.6) | 11(17.4) | | 226(41.5) | 205(49.9) | 43(44.3) | |
| Never married | 22(3.3) | 10(2.4) | 2(2.8) | | 17(2.2) | 13(2.9) | 4(2.9) | |
| BMI[a,b] | | | | 0.101 | | | | <0.001* |
| Underweight | 5(0.8) | 6(1.0) | 1(3.0) | | 6(1.8) | 6(1.5) | 1(1.5) | |
| Overweight | 260(45.0) | 150(36.5) | 22(43.3) | | 183(38.0) | 130(34.6) | 24(28.8) | |
| Obese | 150(28.7) | 143(39.6) | 21(36.4) | | 133(25.0) | 141(33.8) | 44(55.5) | |
| Normal | 150(25.2) | 79(21.9) | 10(16.3) | | 162(35.0) | 99(28.7) | 13(13.7) | |
| Family income/poverty ratio | 3.34 ± 0.07 | 2.94 ± 0.10 | 2.65 ± 0.15 | <0.001* | 3.09 ± 0.10 | 2.53 ± 0.10 | 2.36 ± 0.20 | <0.001* |
| Education[a] | | | | 0.019* | | | | 0.081 |
| Less than 9th grade | 82(8.3) | 72(10.5) | 10(13.6) | | 63(5.4) | 75(11.9) | 16(11.6) | |
| 9–11th grade | 74(10.5) | 57(10.7) | 10(17.0) | | 67(13.0) | 77(19.2) | 20(19.0) | |
| High school grad | 133(22.0) | 91(28.0) | 12(27.0) | | 134(29.4) | 102(29.7) | 23(34.1) | |
| Some college or AA degree | 110(22.0) | 89(26.7) | 14(26.6) | | 124(28.2) | 82(23.5) | 15(20.9) | |
| College graduate or above | 168(37.3) | 74(24.0) | 9(15.8) | | 95(23.5) | 44(15.6) | 9(14.3) | |
| Health insurance status | | | | 0.093 | | | | 0.568 |
| Covered by health insurance | 557(99.4) | 374(98.4) | 52(96.8) | | 471(98.4) | 364(98.2) | 82(99.5) | |
| Not covered by health insurance | 9(0.6) | 8(1.6) | 3(3.2) | | 14(1.6) | 16(1.8) | 1(0.5) | |
| Family/personal medical history | | | | | | | | |
| Hormone replacement therapy | N/A | N/A | N/A | | 232(54.4) | 149(44.1) | 25(41.3) | 0.024* |
| Close relative had asthma | 64(9.7) | 51(10.7) | 6(14.3) | 0.076 | 59(15.0) | 62(15.6) | 14(21.0) | 0.514 |
| Close relative had diabetes | 177(30.5) | 132(35.7) | 24(35.8) | 0.080 | 55(12.8) | 63(18.6) | 15(20.8) | 0.058 |
| Close relative had heart attack | 59(12.0) | 42(12.6) | 8(15.6) | 0.180 | 165(31.1) | 152(39.3) | 33(37.7) | 0.019* |
| Close relative had osteoporosis | 31(6.2) | 22(7.9) | 3(5.5) | 0.272 | 64(16.0) | 41(11.3) | 13(20.5) | 0.114 |
| Hospitalization[a] | | | | <0.001* | | | | <0.001* |
| >3 times | 0(0) | 1(0.3) | 2(5.5) | | 0(0) | 2(0.7) | 2(2.6) | |
| <3 times | 71(14.0) | 80(22.8) | 23(42.9) | | 48(8.7) | 74(19.4) | 26(30.9) | |
| No hospitalization | 496(86.0) | 302(76.8) | 30(51.6) | | 437(91.3) | 304(79.9) | 54(65.8) | |
| Mental health consultation | 6(1.1) | 10(2.7) | 1(2.4) | 0.243 | 6(1.6) | 7(2.0) | 4(4.0) | 0.468 |
| Osteoporosis[a] | 13(2.7) | 21(6.4) | 1(2.8) | 0.007* | 130(29.1) | 84(22.6) | 23(29.8) | 0.243 |

(continued on next page)

Zhang et al. (2018), *PeerJ*, DOI 10.7717/peerj.4326  7/18

| Variables | Male (n = 1005) | | | p-value | Female (n = 948) | | | p-value |
|---|---|---|---|---|---|---|---|---|
| | Non-frail (n = 567) | Pre-frail (n = 383) | Frail (n = 55) | | Non-frail (n = 485) | Pre-frail (n = 380) | Frail (n = 83) | |
| Fracture | 62(12.0) | 50(14.9) | 13(23.3) | 0.107 | 66(13.1) | 54(15.5) | 19(25.1) | 0.114 |
| Steroid usage[a] | 23(5.3) | 18(5.1) | 10(19.2) | 0.001* | 33(8.1) | 20(4.9) | 5(6.7) | 0.388 |

**Notes.**
[a]Summation of percentage is not equal to 100% due to missing data.
[b]Four classifications of BMI are defined as: underweight (BMI $< 18.5$ Kg/m$^2$), normal ($18.5\sim24.9$ Kg/m$^2$), overweight ($25\sim29.9$ Kg/m$^2$), obese ($\geq30.0$ Kg/m$^2$).
*indicates statistical difference between groups, $p < 0.05$.

neoplasms (43.2% vs. 39.9%). Although the numbers are small ($n = 11$) in the frail cluster, the leading cause of death among frail males was heart diseases (41%). A higher mortality rate (8.5%) was found in frail women compared with non-frail (2.5%) and pre-frail (6.4%) women. Among females, most subjects in the non-frail and pre-frail clusters died from other causes (56.6% vs. 24.7%). Although the numbers are small ($n = 7$) in the frail cluster, the leading causes of death among frail females were nephritis, nephrotic syndrome and nephrosis (32.3%) (Table 3).

## Gender-related factors associated with frailty

Results of univariate analysis are summarized in Table 4. Age, marital status, family poverty income ratio, education, health insurance coverage, daily macronutrient intake, high CRP, iron deficiency, physical inactivity, inappropriate sleeping duration, smoking, hospitalization history and history of osteoporosis were associated with frailty in males. After controlling for the effects of other variables, multivariate analysis revealed that six factors were still statistically significant in males, including widowed/divorced/separated (OR = 1.417, 95% CI [1.041–1.929], $p = 0.027$), daily total calorie intake (OR = 1.678, 95% CI [1.072–2.625], $p = 0.023$), physical inactivity (OR = 2.011, 95% CI [1.424–2.838], $p < 0.001$), sleeping hours exceeding 9 h (OR = 1.568, 95% CI [1.005–2.447], $p = 0.047$), smoking (OR = 2.219, 95% CI [1.616–3.046], $p < 0.001$) and hospitalization history (OR = 2.539, 95% CI [1.721–3.745], $p < 0.001$) (Fig. 1A).

In female subjects, univariate analysis revealed that age, obesity, family poverty/income ratio, education, daily macronutrient intake, vitamin D insufficiency, high CRP, moderate alcohol use, physical inactivity, shorter sleep duration, hormone replacement therapy, family history of diabetes and heart attack, and hospitalization history were associated with frailty. After controlling for the effects of other variables, multivariate analysis revealed that only eight factors were still statistically significant, including obesity (OR = 1.685, 95% CI [1.085–2.678], $p = 0.020$), high CRP (OR = 2.735, 95% CI [1.660–4.504], $p < 0.001$), physical inactivity (OR =1.974, 95% CI [1.297–3.004], $p = 0.002$), sleep duration less than 6 h (OR = 1.535, 95% CI [1.078–2.187], $p = 0.018$), family history of diabetes (OR = 1.345, 95% CI [1.024–1.767], $p = 0.033$), family history of heart attack (OR = 2.071, 95% CI [1.535–2.794], $p < 0.001$), and history of hospitalization (OR = 3.182, 95% CI [2.165–4.678], $p < 0.001$). In contrast, higher family poverty/income ratio was associated with lower risk of frailty (OR = 0.828, 95% CI [0.732–0.936], $p = 0.003$) (Fig. 1B).
**Table 2 Subjects' dietary intake, laboratory data and behavioral factors by gender and frailty cluster.**

| Variables | Male ($n = 1,005$) | | | p-value | Female ($n = 948$) | | | p-value |
|---|---|---|---|---|---|---|---|---|
| | Non-frail ($n = 567$) | Pre-frail ($n = 383$) | Frail ($n = 55$) | | Non-frail ($n = 485$) | Pre-frail ($n = 380$) | Frail ($n = 83$) | |
| Daily macronutrient intake[a,b] | | | | | | | | |
| Low total calorie intake | 320(52.7) | 248(62.8) | 38(68.0) | 0.032* | 135(27.7) | 116(30.2) | 13(12.5) | 0.360 |
| Low total protein intake | 111(15.1) | 100(21.6) | 17(34.2) | 0.007* | 92(17.3) | 99(22.0) | 25(30.3) | 0.071 |
| Low total carbohydrate intake | 292(49.2) | 215(54.8) | 34(59.9) | 0.295 | 231(46.5) | 176(42.8) | 42(47.6) | 0.688 |
| Low total fat intake | 120(17.2) | 102(23.2) | 17(33.0) | 0.037* | 100(17.0) | 97(21.5) | 24(30.5) | 0.124 |
| High total saturated fatty acids intake[a] | 394(74.5) | 249(67.7) | 38(70.5) | 0.153 | 346(74.8) | 262(73.8) | 52(68.1) | 0.714 |
| Laboratory exams[a,c] | | | | | | | | |
| Vitamin D insufficiency | 63(7.9) | 61(10.6) | 7(13.7) | 0.333 | 54(9.0) | 72(15.8) | 20(17.2) | 0.016* |
| High CRP | 39(5.7) | 41(9.7) | 7(16.9) | 0.102 | 32(5.6) | 34(8.2) | 14(21.5) | 0.001* |
| Iron deficiency | 58(9.6) | 49(13.0) | 9(20.1) | 0.034* | 13(1.7) | 13(2.3) | 5(5.4) | 0.225 |
| Lifestyle and behavioral factors[a] | | | | | | | | |
| Alcohol consumption | | | | 0.629 | | | | 0.166 |
| Heavy drinker | 7(1.1) | 7(1.4) | 0(0) | | 1(0.2) | 0(0) | 0(0) | |
| Moderate drinker | 446(79.0) | 301(81.4) | 44(81.7) | | 257(57.7) | 162(46.6) | 37(45.6) | |
| Occasional | 111(19.5) | 74(17.0) | 11(18.3) | | 227(42.1) | 216(53.0) | 46(54.4) | |
| Milk consumption | | | | 0.172 | | | | 0.247 |
| Regular | 277(50.8) | 178(47.5) | 30(51.4) | | 215(44.4) | 159(44.4) | 25(29.8) | |
| Never regular | 123(18.6) | 77(20.9) | 4(5.3) | | 122(25.3) | 88(23.6) | 26(25.4) | |
| Sometimes regular | 167(30.6) | 128(31.7) | 21(43.3) | | 148(30.2) | 133(32.0) | 32(44.8) | |
| Vegetarian | 12(1.2) | 7(1.3) | 0(0) | 0.724 | 7(1.0) | 10(2.7) | 4(5.2) | 0.164 |
| Food allergies | 38(7.9) | 21(5.6) | 2(2.5) | 0.437 | 62(12.9) | 50(12.0) | 10(16.1) | 0.732 |
| Water devices use | 188(41.2) | 106(33.5) | 19(38.2) | 0.572 | 162(38.8) | 109(31.9) | 25(38.8) | 0.632 |
| Physical activity (MET minutes) | | | | <0.001* | | | | <0.001* |
| <600 | 213(31.0) | 183(44.0) | 47(87.4) | | 238(44.0) | 255(63.5) | 65(73.8) | |
| >=600 | 354(69.0) | 200(56.0) | 8(12.6) | | 247(56.0) | 125(36.5) | 18(26.2) | |
| Sleeping hours | | | | 0.049* | | | | 0.001* |
| <6 h | 170(27.8) | 144(34.2) | 26(46.1) | | 135(25.7) | 150(34.7) | 34(42.5) | |
| >=9 h | 50(8.4) | 51(13.1) | 6(11.7) | | 39(8.8) | 42(11.4) | 1(0.5) | |
| 7–8 h | 345(63.5) | 188(52.7) | 23(42.2) | | 311(65.5) | 188(54.0) | 48(57.0) | |
| Smoking | 326(54.7) | 276(72.1) | 47(86.8) | <0.001* | 164(35.2) | 114(33.2) | 32(32.9) | 0.779 |

**Notes.**
[a] Summation of percentage is not equal to 100% due to missing data.
[b] According to 2015–2020 Dietary Guidelines for Americans, the recommended daily allowances (RDA) for nutrients are as follows: calorie intake between 2,000∼2,800 Kcal for males and 1,600∼2,200 Kcal for females; protein intake between 50∼245 gm for males and 40∼192.5 gm for females; carbohydrate intake between 225∼455 gm for males and 180∼357.5 gm for females; total fat intake between 44.4∼109 gm for males and 35.5∼85.5 gm for females; saturated fatty acids intake ≤15.6 gm for males and ≤12.3 gm for females.
[c] Normal serum levels are defined as: serum 25-hydroxyvitamin D (25[OH]D) ≥ 40 nmol/L; CRP between 0–1 mg/dL; iron level between 55–160 μg/dL for males and 40–155 μg/dL for females.
*indicates statistical difference between groups, $p < 0.05$.

**Table 3  Causes of death by gender and frailty cluster.**

| Variables | Male (n = 1005) | | | Female (n = 948) | | |
|---|---|---|---|---|---|---|
| | Non-frail (n = 567) | Pre-frail (n = 383) | Frail (n = 55) | Non-frail (n = 485) | Pre-frail (n = 380) | Frail (n = 83) |
| Survival status | | | | | | |
| Alive | 531(94.6) | 346(91.3) | 44(77.5) | 470(97.5) | 356(93.6) | 76(91.5) |
| Deceased | 36(5.4) | 37(8.7) | 11(22.5) | 15(2.5) | 24(6.4) | 7(8.5) |
| Heart diseases | 7(27.2) | 9(19.5) | 4(41.0) | 3(8.1) | 7(24.0) | 0(0) |
| Malignant neoplasms | 17(43.2) | 12(39.9) | 2(16.3) | 4(24.0) | 5(23.0) | 1(11.3) |
| Chronic lower respiratory diseases | 1(2.2) | 0(0) | 2(23.0) | 0(0) | 3(17.0) | 1(17.6) |
| Accidents (unintentional injuries) | 0(0) | 1(1.2) | 0(0) | 0(0) | 0(0) | 0(0) |
| Cerebrovascular diseases | 4(7.6) | 0(0) | 0(0) | 1(9.5) | 0(0) | 1(13.3) |
| Alzheimer disease | 1(1.2) | 0(0) | 0(0) | 0(0) | 1(5.0) | 0(0) |
| Diabetes mellitus | 0(0) | 1(2.8) | 0(0) | 0(0) | 2(6.4) | 0(0) |
| Influenza and pneumonia | 0(0) | 2(7.5) | 0(0) | 1(1.8) | 0(0) | 1(10.7) |
| Nephritis, nephrotic syndrome and nephrosis | 1(3.2) | 0(0) | 0(0) | 0(0) | 0(0) | 2(32.3) |
| All other causes | 5(15.4) | 12(29.1) | 3(19.7) | 6(56.6) | 6(24.7) | 1(14.8) |

# DISCUSSION

## Main findings of this study

In the present study, the prevalence of frailty was higher in females than in males and increased with age in both sexes. Higher mortality rates were also found in men and women categorized as frail, and mortality was higher in men. Although the numbers are small (n = 11) in frail men, the leading cause of death was heart diseases, followed by chronic lower respiratory diseases; non-frail and pre-frail men died of malignant neoplasms. The numbers were also small (n = 7) in frail women, and the leading causes of death were nephritis, nephrotic syndrome and nephrosis, followed by chronic lower respiratory diseases, while non-frail and pre-frail women died from other causes. Independent factors associated with frailty differed between genders. Men who were widowed/divorced/separated, had lower daily total calorie intake, were physically inactive, slept more than 9 h, smoked and were hospitalized previously were more likely to be frail. Frailty was more likely in women who were obese, had elevated CRP indicating inflammation, were physically inactive, slept less than 6 h, were hospitalized previously and had a family history of diabetes or heart attack. Higher family income/poverty ratio was associated with lower risk of frailty in both genders.

## What is already known on this topic

Frailty is understood as an increased risk of adverse health outcomes associated with aging. The causes of frailty are multifactorial, including different types of compromised function and repair processes (*Miller et al., 2017*). The physiological framework that explains frailty involves vulnerabilities associated with aging and chronic disorders; that is, disability in older adults stems mainly from the aging process itself, including unhealthy lifestyles and health disorders and a reduced ability to respond to life's stressors (*Rodriguez-Manas*

**Table 4  Factors associated with frailty: univariate ordinal regression analysis stratified by gender.**

| Univariate | Male | | Female | |
|---|---|---|---|---|
| | OR (95% CI) | *p*-value | OR (95%v CI) | *p*-value |
| **Demographics** | | | | |
| Age | 1.042 (1.018, 1.066) | 0.001* | 1.041 (1.013, 1.07) | 0.006* |
| Race/ethnicity (Ref = Non-Hispanic White) | | | | |
| Mexican American | 1.476 (0.981, 2.22) | 0.062 | 1.118 (0.76, 1.646) | 0.559 |
| Non-Hispanic Black | 1.114 (0.775, 1.601) | 0.556 | 1.519 (0.957, 2.412) | 0.088 |
| Other Hispanic | 0.904 (0.505, 1.62) | 0.736 | 1.094 (0.642, 1.863) | 0.742 |
| Other races | 0.811 (0.369, 1.784) | 0.615 | 1.85 (0.718, 4.767) | 0.167 |
| Marital status (Ref = Married/Living with partner) | | | | |
| Widowed/Divorced/Separated | 1.714 (1.274, 2.305) | 0.001* | 1.31 (0.944, 1.816) | 0.106 |
| Never married | 0.830 (0.368, 1.875) | 0.660 | 1.462 (0.768, 2.783) | 0.248 |
| BMI (Ref = Normal) | | | | |
| Underweight | 2.069 (0.243, 17.61) | 0.539 | 1.207 (0.332, 4.386) | 0.777 |
| Overweight | 1.008 (0.68, 1.494) | 0.968 | 1.224 (0.89, 1.683) | 0.216 |
| Obese | 1.612 (0.994, 2.614) | 0.059 | 2.263 (1.635, 3.132) | <0.001* |
| Family poverty/income ratio | 0.819 (0.739, 0.907) | 0.001* | 0.758 (0.668, 0.86) | <0.001* |
| Education (Ref = High school grad) | | | | |
| Less than 9th grade | 1.065 (0.649, 1.746) | 0.805 | 1.851 (1.157, 2.963) | 0.010* |
| 9–11th grade | 0.905 (0.55, 1.489) | 0.701 | 1.344 (0.82, 2.202) | 0.241 |
| Some college or AA degree | 0.959 (0.562, 1.638) | 0.880 | 0.772 (0.521, 1.146) | 0.199 |
| College graduate or above | 0.488 (0.338, 0.706) | 0.001* | 0.622 (0.373, 1.037) | 0.069 |
| Health insurance coverage | 0.336 (0.128, 0.882) | 0.036* | 1.091 (0.456, 2.609) | 0.838 |
| **Daily macronutrient intake (Ref = higher /recommended)** | | | | |
| Low total calorie intake | 1.559 (1.176, 2.067) | 0.004* | 1.078 (0.804, 1.446) | 0.618 |
| Low total protein intake | 1.756 (1.326, 2.325) | <0.001* | 1.523 (1.118, 2.075) | 0.013* |
| Low total carbohydrate intake | 1.285 (0.973, 1.698) | 0.097 | 0.919 (0.663, 1.274) | 0.617 |
| Low total fat intake | 1.591 (1.132, 2.235) | 0.014* | 1.541 (1.101, 2.158) | 0.018* |
| High saturated fatty acids intake | 0.729 (0.565, 0.94) | 0.019* | 0.867 (0.598, 1.256) | 0.458 |
| **Laboratory exams (Ref = Normal)** | | | | |
| Vitamin D insufficiency | 1.489 (0.865, 2.566) | 0.162 | 1.889 (1.266, 2.818) | 0.003* |
| High CRP | 2.053 (1.081, 3.901) | 0.038* | 2.492 (1.5, 4.141) | 0.002* |
| Iron deficiency | 1.614 (1.081, 2.411) | 0.019* | 2.002 (0.81, 4.944) | 0.163 |
| **Lifestyle and behavioral factors** | | | | |
| Alcohol consumption (Ref = Occasional) | | | | |
| Heavy drinker | 1.181 (0.347, 4.021) | 0.781 | N/A[a] | |
| Moderate drinker | 1.164 (0.909, 1.491) | 0.239 | 0.648 (0.437, 0.962) | 0.040* |
| Milk consumption (Ref = Regular) | | | | |
| Never regular | 1.023 (0.685, 1.528) | 0.910 | 1.053 (0.784, 1.414) | 0.732 |
| Sometimes regular | 1.164 (0.82, 1.652) | 0.406 | 1.308 (0.83, 2.06) | 0.255 |

**Table 4** (*continued*)

| Univariate | Male | | Female | |
|---|---|---|---|---|
| | OR (95% CI) | *p*-value | OR (95%v CI) | *p*-value |
| Vegetarian (Ref = No) | 0.892 (0.318, 2.501) | 0.829 | 3.174 (0. 696, 14.477) | 0.136 |
| Food allergies (Ref = Yes) | 1.586 (0.762, 3.301) | 0.221 | 0.963 (0.518, 1.792) | 0.908 |
| Water devices use (Ref = No) | 0.754 (0.511, 1.110) | 0.152 | 0.821 (0.573, 1.177) | 0.283 |
| Physical inactivity (METs <600) (Ref: METs > = 600) | 2.398 (1.749, 3.288) | <0.001[*] | 2.416 (1.573, 3.709) | <0.001[*] |
| Sleeping hours (Ref = 7–8 h) | | | | |
| <6 h | 1.625 (1.205, 2.19) | 0.003[*] | 1.677 (1.22, 2.307) | 0.003[*] |
| >=9 h | 1.854 (1.283, 2.679) | 0.002[*] | 1.094 (0.699, 1.712) | 0.679 |
| Smoker (Ref = Non-smoker) | 2.394 (1.775, 3.23) | <0.001[*] | 0.914 (0.721, 1.159) | 0.464 |
| Hormone replacement therapy (Ref = No HRT) | N/A | | 0.663 (0.506, 0.87) | 0.006[*] |
| Close relative had asthma (Ref = No) | 1.226 (0.705, 2.133) | 0.479 | 1.191 (0.782, 1.814) | 0.427 |
| Close relative had diabetes (Ref = No) | 1.289 (0.975, 1.705) | 0.084 | 1.412 (1.079, 1.846) | 0.017[*] |
| Close relative had heart attack (Ref = No) | 1.141 (0.657, 1.981) | 0.646 | 1.597 (1.231, 2.073) | 0.001[*] |
| Close relative had osteoporosis (Ref = No) | 1.196 (0.609, 2.349) | 0.600 | 0.847 (0.621, 1.155) | 0.324 |
| Hospitalization (Ref = No hospital utilization) | 2.391 (1.652, 3.46) | <0.001[*] | 3.118 (2.236, 4.349) | <0.001[*] |
| Mental health consultation (Ref = No consultation) | 2.274 (0.837, 6.177) | 0.104 | 1.647 (0.654, 4.153) | 0.322 |
| Osteoporosis (Ref = No) | 2.005 (1.032, 3.896) | 0.038[*] | 0.818 (0.548, 1.222) | 0.343 |
| Fracture (Ref = No) | 1.441 (0.949, 2.188) | 0.102 | 1.496 (0.906, 2.47) | 0.134 |
| Steroid usage (Ref = No) | 1.571 (0.79, 3.122) | 0.247 | 0.653 (0.352, 1.212) | 0.197 |

**Notes.**

Abbreviations:: CI, confidence interval; MET, metabolic equivalent task; N/A, not available; OR, odds ratio; Ref, reference.

[a] Only one heavy drinker was found among females, so subject was re-classified into moderate drinker group for analysis.

*indicates statistical difference between groups, *p* < 0.05.

& *Fried, 2015*). Measuring frailty is most often done using the frailty phenotype model and the frailty index. The FRAIL scale (*Morley, Malmstrom & Miller, 2012*) used in the present study incorporates aspects of both but discriminates frailty at the lower prevalence levels (*Blodgett et al., 2015b*). Using the five-item self-reported questionnaire (FRAIL scale) (*Morley, Malmstrom & Miller, 2012*), has been shown to estimate frailty prevalence accurately in community-dwelling older adults (*Yamada & Arai, 2015*). Consistent with our results, the prevalence of frailty is higher among women and increases with age in both genders (*Rodriguez-Manas & Fried, 2015*). Development of frailty in women is influenced by deficits of various hormones during aging and increased inflammatory states (*Eichholzer et al., 2013*). Overlap in frailty, disability, and comorbidities is a combination of causes contributing to mortality, and overlap is associated especially with greater frailty (*Theou et al., 2012*). Prevention is possible. A research team conducting a 10-year community intervention for frailty prevention among older adults (>65) used a public health approach with community consensus to develop a health education program that effectively promoted healthy aging; the success of the program relied on comprehensive assessment of geriatric individuals and improving self-care ability (*Shinkai et al., 2016*).

## What this study adds

We found common predictive factors for frailty among older adults of both genders, including more frequent previous hospitalizations and physical inactivity, consistent

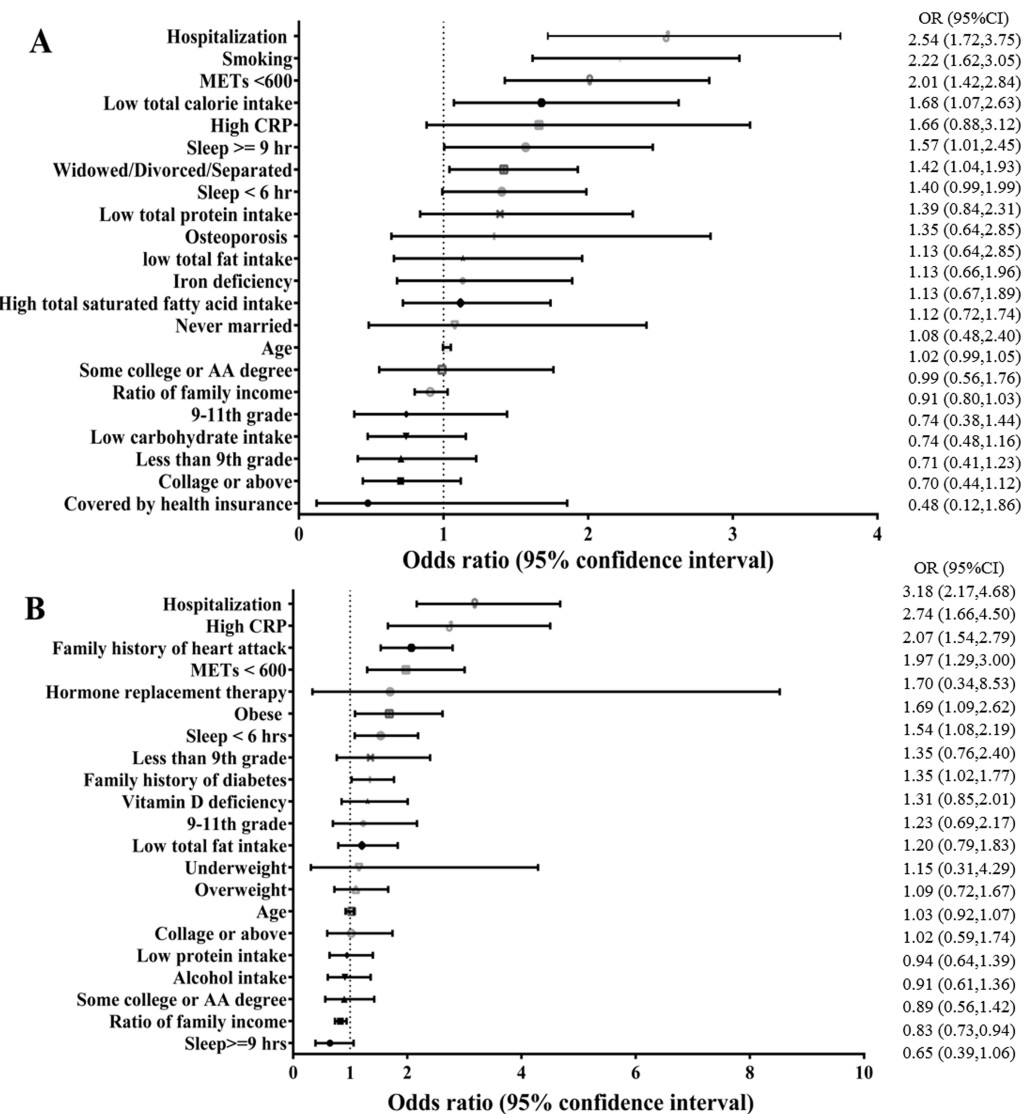

**Figure 1 Risk factors for geriatric frailty: multivariable regression analysis in males (A) and females (B).**

with results of other NHANES studies in which self-reported higher levels of illness and sedentary behavior were directly associated with frailty (*Blodgett et al., 2015a*). Notably, non-frail older adults of both genders had a higher family income/poverty ratio, which gradually decreased toward the frail group, indicating that greater financial security helps people take better care of their health. However, independent risk factors for frailty showed that, for men, being widowed, divorced, or separated increased the likelihood of frailty along with low daily total calorie intake, sleeping too much, and smoking, suggesting that men living alone take poor care of their nutritional needs and persist in harmful lifestyle habits. Correspondingly, better diet quality is associated with lower odds of developing frailty (*Chan, Leung & Woo, 2015*). In women, obesity, inflammation

(high CRP), insufficient sleep, and family history of diabetes or heart attack increased the likelihood of frailty, suggesting a high-stress burden. Our study population had more males but frailty was higher among women. Although women live longer than men, our results agree with other reports, showing that women tend to have poorer health status. Even in developed countries, the environment is more adverse for women, and lifestyle factors may increase women's vulnerability to stochastic subcellular events that increase recovery time (*Hubbard & Rockwood, 2015*). In our study, this was evident in the mean age of frail females, which was 73.7 years, younger than the 75.4 years of frail males. However, the mortality of frail men exceeded that of women. Major causes of death among frail women were chronic kidney diseases, of which all stages are associated with frailty (*Wilhelm-Leen et al., 2009*), and chronic lower respiratory diseases. These leading causes of death are longer, slower disease processes that contribute to progressive debilitation, helping to explain why mortality among frail women is lower than in frail males.

## Strengths and limitations

The present study is strengthened by the use of NHANES, which is a well-known and often used national health survey and one of the few population-based surveys that include validated examination measures, biological specimen collection, and limited measures of health status. The analysis in the present study was conducted in a nationally representative sample, allowing our results to be generalized to the entire U.S. adult population. This study also has some limitations, including that it used cross-sectional analysis, which limits inferences regarding causality. Also, the FRAIL scale was used for evaluating frailty among NHANES participants, and it is not known whether other scales may have produced different results. Although frailty is considered a predictor of all-cause mortality (*Kulmala, Nykänen & Hartikainen, 2014*; *Lin et al., 2016*), only a small number of participants were included in the deceased/mortality groups categorized by frailty, which limited our findings and precluded making comparisons relative to cause of death as reported in the above-mentioned studies. A major limitation is that the sample is not geographically representative of the United States, even while it is demographically representative; because the two NHANES teams could only visit 16 places each year, achieving a good geographic spread was not possible. NHANES data also do not represent observed changes over time. In-person interview data (by questionnaire) were based on self-reports and subject to recall problems, misunderstanding of questions, and various other factors. NHANES is US data (including representative proportions of different ethnic groups) and needs to be validated in other countries.

## CONCLUSIONS

Older adults categorized as frail have higher mortality rates than those who are pre-frail or non-frail. Factors associated with frailty differ by gender, with higher frailty prevalence in females and higher mortality in males. Although numbers are small in the present study, the leading causes of death among frail older adults are heart diseases in males and chronic kidney diseases in females. Gender-associated factors for frailty identified in this study

may be useful in evaluating frailty and guiding development of public health measures for prevention.

## ACKNOWLEDGEMENTS

The authors wish to thank the US National Center for Health Statistics (NCHS) for creating the National Health and Nutrition Examination Survey and making the NHANES data available to researchers. We also acknowledge that interpreting and reporting these data are the sole responsibility of the authors.

### Funding

This study was supported by The Zhejiang Medical and Health Science and Technology Project (No. 2017KY335) and The Construction Project of National Key Clinical Geriatrics Department. The funders had no role in study design, data collection and analysis, decision to publish, or preparation of the manuscript.

### Grant Disclosures

The following grant information was disclosed by the authors:
The Zhejiang Medical and Health Science and Technology Project: No. 2017KY335.
The Construction Project of National Key Clinical Geriatrics Department.

### Competing Interests

The authors declare there are no competing interests.

### Author Contributions

- Qin Zhang conceived and designed the experiments, performed the experiments, analyzed the data, wrote the paper.
- Huanyu Guo contributed reagents/materials/analysis tools.
- Haifeng Gu analyzed the data, prepared figures and/or tables, reviewed drafts of the paper.
- Xiaohong Zhao conceived and designed the experiments, wrote the paper.

### Data Availability

  The raw data has been provided as Supplemental File.

### Supplemental Information

Supplemental information for this article can be found online at http://dx.doi.org/10.7717/peerj.4326#supplemental-information.

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
