# Peer review of "Gender-associated factors for frailty and their impact on hospitalization and mortality among community-dwelling older adults: a cross-sectional population-based study"

_PeerJ, doi:10.7717/peerj.4326_

## Round 0.1 · original submission · Major Revisions

Dear authors,

I have carefully read the comments of the reviewers and both of them have indicated your manuscript has scientific merit if all their changes are applied. Taking into account that there are a lot of modifications, my decision is MAJOR REVISION.

With respect and warm regards,
Dr Palazón-Bru (academic editor for PeerJ)

·

Basic reporting

I commend the authors for this very interesting and good quality paper investigating the gender-associated risk factors for geriatric frailty and their impact on hospitalization and mortality among community-dwelling older adults. For the most part this paper is well written and well structured. The background to the research question and aims are quite well described and the manuscript progresses well with clearly described results and a thoughtful discussion. There is good logical development throughout the manuscript with careful interpretation of the results and the implications of the findings. The research aims could be emphasised by the inclusion of additional information in the introduction which I describe in more detail in the General Comments below . There are also some grammatical issues which I have listed in the General Comments below.

Experimental design

The primary research question and aims fall within the scope of this journal. The research questions are well defined, relevant & meaningful. The paper clearly addresses a knowledge gap in relation to identifying the gender differences that exist when it comes to risk factors for frailty. Indeed the importance of gender differences in relation to disease risk is a very current and important topic of interest in clinical and epidemiological research
The study design is appropriate and robust enough to address the research question and the statistical methodology applied is also valid and appropriate.
The study uses a well described dataset from the highly reputable NHANES study from the USA. The study population, outcomes of interest and risk factors investigated are well described and defined with sufficient detail and information to replicate the analyses.

Validity of the findings

A stated above the study, design, data and statistical approach of this study are robust sound, & controlled. However, there are small number of participants within the deceased/mortality groups categorised by frailty. Therefore I would encourage the authors to acknowledge this in the Limitations of the Discussion section and indicate the replication of the findings in relation to cause of death in other studies
The conclusion are well stated, are a logical implication of the study results and link directly to the original research question.
The authors are thoughtful in their discussion and conclusion and do not to over-speculate, which is to be commended.

Additional comments

Major Issues
Introduction
1. The authors provide a good background to the importance of frailty and how it is measured. Examining gender differences in risk factors for frailty is the main aim of this paper. However, what is known/unknown about gender differences for frailty and frailty risk factors is not addressed. A short paragraph describing what is known/unknown about gender differences for frailty and frailty risk factors would improve the introduction and would highlight why their research question and aim of the study is important. Some of the introduction describing the different frailty measurements could be sacrificed to include information on what is known/unknown about gender differences in frailty and risk factors for frailty.
Patients and Methods
1. Lines 100-102: Please could the authors refer to the NHANES website and clarify that ethical approval and participant written informed consent was obtained prior to data collection by the NHANES study. Therefore, no further ethical approval and informed consent was required to perform the secondary analyses undertaken in this manuscript.
2. Line 105: Please provide the mean age with age range, also percentage female in the study population of the community-dwelling geriatric subjects.
3. Line 115-116: According to the Morley FRAIL scale, pre-frail individuals are classified as exhibiting 1-2 of the five FRAIL criteria specifically, not “individuals at risk for frailty who fulfil some, but not all, frailty criteria”. The authors should briefly clarify that the FRAIL scale scores of 0, 1-2, or 3-5 classify individuals as robust, pre-frail and frail respectively.
4. Lines 119-121: When describing the mortality data, please indicate how long the follow-up period was for the mortality data in years, or what was the cut-off date for the mortality data.
5. Lines 126: “disease associations” does not seem like an appropriate group title, perhaps “health and medical conditions” would be a better way to describe the variables (self-reported osteoporosis, fracture, prednisone or cortisone use, mental health consultation). Also hormone replacement therapy is more appropriately included with the “health and medical conditions” variables rather than lifestyle factors. This how these variables are described and grouped in Lines 206-208 of the Patients and Methods.
6. Lines 219-222: In the Statistical Analysis section the authors apply the correct statistical tests and provide the commands from the SAS statistical software package. It would be useful if they included the actual name of the statistical test they applied. For example, Weighted linear regression analyses (PROC SURVEYREG) were used to test differences in continuous characteristics between participants clustered according to health status levels of robust, pre-frail, and frail. This should be done for PROC SURVEYFREQ and PROC SURVEYLOG also.
Results
1. It would be helpful if the authors could provide a table comparing some of the characteristics of the study group (n=1953) and the group of participants with missing data (n=1126). This would allow some understanding of whether the missing participants are missing at random, or due to some other effect e.g. Are the missing participant group older, more male, in poorer health etc. Or are there no real differences between the study group and the missing data group.
2. Lines 223-236: for the males, please indicate the direction of effect in the text for each variable significantly higher/lower in the pre-frail and frail.
For example: “However compared to robust, pre-frail and frail males were significantly older, more likely to be widowed/divorced, had lower family poverty income ratios, had lower levels of educational attainment, etc.
3. 241-244: Same for females, please include the direction of effect in the text for each variable significantly higher/lower in the pre-frail and frail.
4. Lines 247-255: Please state that although the numbers are small (n=11), the leading cause of death among frail males was Y condition. Also please state the same for the frail females (n=7).
5. Lines 264-267: Please do not list the Odds Ratios and p-values in one sentence. Please put the Odds Ratios and p-values beside the corresponding variables in lines 261-263. This was done correctly for the females in lines 273-280.
Table 1
1. Please include frailty cluster and gender in the title of the Table 1, as you have done for Table 2.
“Subjects’ demographic and anthropometric characteristics and personal/family medical history by gender and frailty cluster”
2. The p-value is missing for Family income/poverty ratio by frailty category among females.
Table 2.
1. Please include frailty cluster in the title of the Table 2
“Subjects’ dietary intake, laboratory data and behavioral factors by gender and frailty cluster!
Table 3.
1. Please include gender and frailty cluster in the title of the Table 3
“Causes of death by gender and frailty cluster”
Discussion
1. Lines 286-289 Please also indicate the leading cause of death among frail males and frail females, but also stating that the number were very low.
2. Lines 300-303: The point the authors are making here is important and correct. However, this sentence is too long and difficult to fully understand. Please rewrite as 2 or 3 shorter sentences. This will improve understanding and emphasise the point the authors are making about the etiology of frailty.
3. Line 304: Please rephrase for clarity “frailty phenotype model and the frailty index, the FRAIL scale (Morley et al. 2012) used in this study incorporates aspects of both but discriminates frailty at a lower prevalence level.
4. Lines 306-308: Please remove the first part of this sentence “Prediction of frailty is possible u Using the 5-item self-reported questionnaire (FRAIL scale) (Morley et al. 2012), has been shown to estimate frailty prevalence accurately in community-dwelling older adults (Yamada & Arai 2015)”
5. There are a small number of participants within the deceased/mortality groups categorised by frailty cluster. Therefore I would encourage the authors to acknowledge this in the Limitations of the Discussion section, and indicate that replication of the findings in relation to cause of death in other studies is warrented.
Minor Issues
7. Short running title: Please include a reference to gender as it is gender differences that are fundamental to this paper e.g. gender-related risks for geriatric frailty
8. Keywords: Please include gender, as gender differences are fundamental to this paper.
9. Key messages: it may be worth mentioning that there are differences in the gender-related risk factors as well as the commonalities that they highlight. I find the commonalities (chronic lower respiratory diseases) and differences in the leading cause of death causes of death interesting also, though not directly linked to frailty.
10. Line 67: Please replace “At the same time as…” with “While…. “ or with “Concomitantly frailty in older adults is associated with increased symptoms, complex diagnoses, and diminishing tolerance for medical interventions”.
11. Line 83: Please replace “Managing geriatric patients must be able to diagnose frailty…” with “Managing geriatric patients requires the ability to diagnose frailty….”
12. Line 85: Include a space between the end of the sentence and the reference. and death (Miller et al. 2017).
13. Lines 95-96: Please replace “of USA” with “in the USA”.
14. Line 106: (Morley et al 2012) is an incorrect reference, should this be Miller et al 2017 reference?
15. Lines 119: The authors provided mortality data for all participants in the Results and Table 3, not just the frail. Please remove the word frail from this line.
16. Line 132: Please replace “collected” with “analysed”, the data was not collected for males and females separately but was analysed by the authors separate

·

Basic reporting

The topic is of significant interest but English needs significant improvement, The references are relevant, as the article structure. I am not sure about the hypothesis is good and supported by the results. There is nothing in your results that suggest anything about the help to clinicians to assess frailty and reduce health outcomes. You have to rethink this part of your study. The results are, however, of some interest (see my general/specific comments below) so I don;t want to repeat here.

Experimental design

Research question is not well defined (as I also said about the hypothesis) but still I think it can be rewritten as the results are of some interest.

Validity of the findings

Conclusions should be worked out - for now they are purely speculative but this can be improved. You have your finding and just have to think how to summarize them in one-two sentences conclusion.

Additional comments

In the manuscript, you analyzed the associations between many different health-related factors with frailty operationalized using the FRAIL scale. I am not sure that you can call them the risk factors for frailty (I am not also sure about the term “geriatric frailty” is good, frailty is predominantly applied for geriatric populations. The manuscript is well written (in some parts) but needs significant rewording in the other parts. Even the first sentence in the introductions needs some work. The other example is in lines 300, you might say “frailty is the biological phenomenon” not “biological framework”. The other example is in lines 339-341. The list of such places can be extended. I suggest to show your manuscript to a native English speaker- it would help.

Discussion in the abstract is too short, looks more like conclusion, although to be considered as conclusion it must be more substantial. I completely disagree with what is in Conclusion now- this is too speculative and it might fit for discussion not for conclusions.
The term “robust” is not a good, John Morley (the author of the FRAIL scale) called this category as “healthy” – I would suggest just “non-frail”- as it is not clear how healthy or robust they are.

Statistical analysis section. It is not clear for me how the sex-differences were assessed: by categories of frailty or overall? Reference to the SAS PROC (as any subroutine) is not needed. Sufficient to say that SAS was used for the analyses.

In Tables 1 & 2, just a few variables showed statistically significant difference between the categories of frailty by sex but not all results are intuitively clear. For example, the vitamin D insufficiency is found significance while Iron deficiency was not despite it looks otherwise from the numbers (that impression might be wrong) that is why is important to have more details in the methods section.

Table 3 shows that the death rate increased by the category of severity in both sexes.

Table 4 shows generally known significant relationships of frailty with some factors such as age, obesity, income, and some macronutrient intake or vitamin D insufficiency, physical inactivity and sleeplessness.

I suggest to put the Strength and Limitations section before conclusions- Conclusions means the final words. We have also to keep in mind that the results are obtained for the FRAIL scale, the other scale might not necessarily to give the same results. It could be said in this (limitations) section or could be provided some evidence that the other scales also identify these factors to be associated with frailty. The latter would be more interesting and important. If that is the case, it indicates that frailty as a phenomena is robust (here we can use this word) and however defined has the same associated factors (e.g., age, obesity, low exercise, etc.).

Figure 1 would be better to have the same order in both panels or perhaps alternatively to order them from the highest to lowest.

---

## Round 0.2 · Minor Revisions

Dear authors,

Your manuscript has high standards to be published in PeerJ, but before this, you should perform some minor changes to the text.

With respect and warm regards,
Dr Palazón-Bru (academic editor for PeerJ)

·

Basic reporting

The article is much improves following major revision and re-submission.
It now meets the criteria for basic reporting, although I would recommend one final review by a native English speaker to correct any minor grammatical issues.

Experimental design

The experimental design is valid and additional work on the aims of the study and the methods and analyses have improved the quality and clarity of the study.

Validity of the findings

The study, design, data and statistical approach of this study are robust sound, & controlled. The conclusion are well stated, are a logical implication of the study results and link directly to the original research question.
The authors are thoughtful in their discussion and conclusion and do not to over-speculate, which is to be commended.

Additional comments

I believe the authors have made every effort and have addressed sufficiently the issues raised by the reviewers.
I might just recommend one final review by a native English speaker to address any final minor grammatical issues.

·

Basic reporting

The authors responded to my questions/ comments and generally I am satisfied with their revised version of the manuscript. The manuscript satisfy the requirements to meet with the standards in terms of the background, the structure and the results.

Experimental design

Methods were described in sufficient details that are quite appropriate in the field.The results contribute to the area of frailty and aging.

Validity of the findings

Validity of the finding are not in any doubt. The results are not unexpected but still contribute to the body of knowledge in the research on aging and frailty.

Additional comments

The authors responded to my questions and made corrections according to my suggestions/ comment, One minor thing. The word "benchmark" in respect to the NHANES (see line 348) is not the best. I would suggest to change it to, for example "a well known" or "often used", or something like that..

---

## Round 0.3 · accepted · Accept

Dear authors,

I am happy to inform you that your paper has been accepted for publication in PeerJ.

Congratulations!

With respect and warm regards,
Dr Palazón-Bru (academic editor for PeerJ)

·

Basic reporting

I have no problems with the manuscript as the authors accounted to my comments/ concerns

Experimental design

There is no changes from the previous review- I have no problem with what called here experimental design

Validity of the findings

The findings are valid

Additional comments

The authors responded well to my questions/ concerns